# Simultaneous Determination of Pyrethroid Insecticides in Foods of Animal Origins Using the Modified QuEChERS Method and Gas Chromatography-Mass Spectrometry

**DOI:** 10.3390/foods11223634

**Published:** 2022-11-14

**Authors:** Byung Joon Kim, Seung-Hyun Yang, Hoon Choi

**Affiliations:** 1Hansalim Agro-Food Analysis Center, Hankyong National University Industry Academic Cooperation Foundation, Suwon 16500, Korea; 2Healthcare Advanced Chemical Research Institute, Environmental Toxicology & Chemistry Center, Hwasun-gun 58141, Korea; 3Department of Life and Environmental Sciences, Wonkwang University, Iksan 54538, Korea

**Keywords:** acetonitrile, GC-MS/MS, foods of animal origins, pyrethroid, QuEChERS

## Abstract

Pyrethroid insecticides are used in agriculture to treat parasites in livestock. This study developed a simultaneous residue analysis method to measure seventeen pyrethroid insecticides in foods of animal origin, including beef, pork, chicken, milk, and eggs. The method, which comprises instrumental analysis using gas chromatography-tandem mass spectrometry and a modified QuEChERS (quick, easy, cheap, effective, rugged, and safe) method for pretreatment, was optimized to verify the applicability of the method. A mixture of acetonitrile, ethyl acetate, and original salt (MgSO_4_ 4 g, NaCl 1 g) was used as the extraction solvent and salt. MgSO_4_ (150 mg) primary secondary amine (25 mg) and graphitized carbon black (25 mg) were selected for dispersive solid phase extraction (d-SPE). The method limit of quantitation was 0.01 mg/L, and the linearity of the matrix-matched calibration curves was reasonable (R2 > 0.99). Recovery tests were performed at three concentrations (LOQ, 10 LOQ, and 50 LOQ). Good recoveries (75.2109.8%) and reproducibility (coefficient of variation <10%) were obtained. The matrix effects were in the range of –35.8 to 56.0%. The established method was fully validated and can be used as an official analytical method for quantifying pyrethroid insecticides in animal commodities.

## 1. Introduction

Pyrethroid insecticides (PIs), derived from pyrethrum flowers, are used worldwide. However, in the early stages after application, PIs are vulnerable to photolysis [1]. Therefore, synthetic pyrethroids that can withstand photolysis are being developed [2]. Generally, pyrethroids exert an insecticidal effect by interfering with the sodium channel of the nervous system. Most PIs have many stereoisomers because the chiral center of the carbon is connected to the structure’s cyclopropane ring and nitrile group. Pyrethroids are insecticides used to treat livestock parasitic diseases [3]. PIs such as cyfluthrin, cypermethrin, deltamethrin, and permethrin are used as veterinary drugs in the Republic of Korea [4]. Deltamethrin and cypermethrin are also widely used to control ticks and pests in agriculture [5]. Moreover, PIs, including arinathrin, bifenthrin, cyhalothrin, etofenprox, fenpropathrin, fluvalinate, flucythrinate, bioresmethrin, silafluofen, and tefluthrin, have the same insecticidal activity and can be used on livestock farms. As mentioned above, livestock can easily be exposed to pyrethroids to control parasitic diseases. Therefore, it is necessary to accurately analyze all the PIs in livestock.

PIs are typically detected by gas chromatography (GC) because of their polarity and boiling point [6]. In addition, the heat stability of pyrethroids prevents their degradation or isomerization [7]. PIs contain halogen and nitrile groups in their structure, leading pyrethroids to be analyzed using GC electron capture detector (GC–ECD) analysis [8]. Gas Chromatography-tandem mass spectrometry (GC–MS/MS) can be used to exclude potential interfering peaks from the sample [9]. This means that GC–MS/MS has better selectivity and sensitivity, leading to a much lower limit of quantification (LOQ) [10,11,12]. PIs are extracted using various solvents such as acetonitrile, acetone, and methanol [1]. Sometimes a combination of two different solvents is used. The QuEChERS (quick, easy, cheap, effective, rugged, and safe) method is generally applied for pretreatment [13,14]. The QuEChERS method typically involves the use of acetonitrile as an extraction solvent, and this solvent is suitable for liquid chromatography-tandem mass spectrometry (LC–MS/MS) and GC–MS/MS [15]. Moreover, the extraction solvent can combine acetonitrile and other solvents, such as ethyl acetate, hexane, and dichloromethane, which are used for high-fat samples [14,16,17,18]. Florisil and silica SPE cartridges are typically used for cleanup [1]. There are various dispersive solid-phase extraction (d-SPE) sorbents that can be applied to a number of food matrices for the QuEChERS cleanup [15]. Research has shown that QuEChERS extraction can be combined with SPE cartridges for better cleanup [19]. Table 1 shows the extraction solvent and pretreatment used in methodologies including pyrethroid insecticide.

To the best of our knowledge, no analytical method has been developed to simultaneously analyze PIs, pesticides, and veterinary drugs in animal-derived foods. Hence, the main purpose of this study was to develop a reliable analysis method for foods of animal origin for seventeen PIs: acrinathrin, bifenthrin, cyfluthrin, cyhalothrin, cypermethrin, deltamethrin (tralomethrin), etofenprox, fenpropathrin, fenvalerate/esfenvalerate, tau-fluvalinate, flucythrinate, halfenprox, permethrin, phenothrin, resmethrin/bioresmethrin, silafluofen, and tefluthrin. The analysis method was optimized for efficiency using comparative experiments. 

## 2. Materials and Methods

### 2.1. Chemicals and Reagents

Acrinathrin (≥98.0%), bifenthrin (≥98.0%), cycloprothrin (≥98.0%), cyfluthrin (≥95.0%), cyhalothrin (≥98.0%), cypermethrin (≥90.0%), deltamethrin (tralomethrin) (≥98.0%), etofenprox (≥98.0%), fenpropathrin (≥98.0%), fenvalerate/esfenvalerate (≥98.0%), tau-fluvalinate (≥95.0%), flucythrinate (≥95.0%), halfenprox (≥98.0%), permethrin (≥90.0%), phenothrin (≥90.0%), and tefluthrin (≥95.0%) were purchased from Sigma-Aldricsentence (St. Louis, MO, USA). Resmethrin/bioresmethrin (1000 mg/L in acetonitrile) and silafluofen (1000 mg/L in acetonitrile) were obtained from the Ministry of Food and Drug Safety (Cheongju, Korea). Acetonitrile, acetone, ethyl acetate, and distilled water (HPLC grade) were purchased from Fisher Scientific (Seoul, Korea). QuEChERS extraction salt original, EN, AOAC packages, and d-SPE (MgSO_4_), primary secondary amine (PSA), graphitized carbon black (GCB and C18) were obtained from Restock (Bellefonte, PA, USA). 

### 2.2. Instrumental Conditions

The MS conditions were optimized using an Agilent 7890B coupled with an Agilent 7000C MS. A standard solvent mixture (5 mg/L) was injected for multiple reaction monitoring (MRM). A ZB-SemiVolatiles (0.25 mm i.d. × 30 m, 0.25 µm film thickness, Phenomenex, Torrance, CA, USA) column was used. Helium (≥99.999%) was used as the carrier gas at a 1.0 mL/min flow rate. Various splitless pulsed pressure modes were examined (nonpulsed, 10, 20, 35, and 40 psi), and 2 μL was injected. The oven temperature followed a gradient and was programmed as follows: 90 °C for 3 min, increased to 180 °C at 20 °C/min, held for 3 min, increased to 260 °C at 15 °C/min, held for 2 min, increased to 300 °C at 10 °C/min, and held for 5 min. The total runtime was 26.83 min. Ionization was performed using electron ionization (EI), and the electron voltage was 70 eV. The source temperature was 230 °C, and the MS transfer line temperature was 280 °C. Nitrogen (≥99.999%) was used as the collision gas.

### 2.3. Sample Preparation: Extraction & Purification

To optimize the extraction conditions, the extraction solvents and salts were compared. Extraction solvent comparisons were performed by fortifying 0.01 mg/L of the standard mixture in 5 g of sample. Acetonitrile, acetone, and ethyl acetate/acetonitrile (1:9, 2:8, and 3:7, *v*/*v*, 20 mL) were compared to optimize the extraction solvent.

Three different extraction salts were used for the extraction salt optimization: (A) Original salt—MgSO_4_ 4 g and NaCl 1 g, (B) AOAC salt—MgSO_4_ 6 g and NaSO_4_ 1.5 g, and (C) EN salt—MgSO_4_ 4 g, NaCl 1 g, disodium hydrogencitrate sesquihydrate (HOC(COOH)(CH_2_COONa)_2_·1.5H_2_O) 1 g, and trisodium citrate dihydrate (HOC(COONa)(CH_2_COONa)_2_·2H_2_O) 0.5 g. For the comparison of extraction salt, 0.01 mg/L of the standard mixture was added to 5 g of sample. 

For the clean-up procedure, four different types of d-SPE were compared: (A) d-SPE I—MgSO_4_ 150 mg, and PSA 25 mg, (B) d-SPE II—MgSO_4_ 150 mg, PSA 25 mg, and C18 25 mg, (C) d-SPE III—MgSO_4_ 150 mg, PSA 25 mg, and graphitized carbon black (GCB) 25 mg, and (D) d-SPE IV—MgSO_4_ 150 mg and C18 25 mg. For the comparison, 0.01 mg/L standard mixture was fortified.

### 2.4. Method Validation

Method validation was performed according to SANTE/11945/2015 [20]. For the method validation, 0.01 mg/L, 0.1 mg/L, and 0.5 mg/L fortified levels were used. The recovery (%) and coefficient of variation (CV, %) were calculated at each fortified level. Hence, enhancement and suppression resulted in signals based on matrix components, and matrix-matched standards were used for quantification. A five-point matrix-matched standard calibration was used (5, 10, 15, 20, and 30 μg/L). The matrix effects (MEs) were calculated using the following equation: ME (%) = (S_matrix_/S_solvent_ − 1) × 100, where S_matrix_ and S_solvent_ are the slopes of the matrix-matched standard curve and standard curve in acetonitrile, respectively.

## 3. Results and Discussion

### 3.1. Multiple Reaction Monitoring Optimization

PIs have been widely analyzed using liquid chromatography (LC) and GC [21]. Owing to the presence of a halogen compound in the pyrethroid structure, electron capture detection has been used to analyze PIs by GC [22]. In addition, some reports on the analysis of PIs used GC–MS/MS [3,6,16,23]. Because the molecular weights of the target compounds were below 600 and the compounds were relatively non-polar, they were suitable for GC–MS/MS analysis. Therefore, GC–MS/MS was chosen for the analysis. MRM is one of the most popular analytical methods for GC–MS/MS. Precursor and product ions were selected for multiple reaction monitoring and used for quantification and qualification. 

To optimize the multiple reaction monitoring conditions, 5 mg/L of the standard mixture dissolved in acetonitrile was used. A full scan was obtained within a spectral mass range of 10–500 m/z. Precursor ions were selected based on the sensitivity and selectivity of each compound. A product ion scan was performed based on the precursor ion chosen by applying various ranges of collision-induced dissociation energy (1–60 eV). Quantifier and qualifier ions were selected based on their selectivity and sensitivity. Cycloprothrin was excluded at this step owing to its poor sensitivity. However, tralomethrin was transformed into deltamethrin by thermal decomposition; therefore, tralomethrin was quantified as deltamethrin. As a result, 17 PIs were selected after multiple reaction-monitoring optimizations (Table 2).

### 3.2. Instrumental Conditions

MS columns have been used in several optimized GC–MS/MS methods [6,10,11]. The columns used in those studies had the same MS column size as in this experiment. For the most suitable selectivity and sensitivity, a SemiVolatiles (0.25 mm i.d., ×30 m, 0.25 µm film thickness, Phenomenex, Torrance, CA, USA) column was used. The pulsed pressure injection mode can increase the peak area, leading to a much lower limit of quantitation (LOQ) [24]. The injection volume was 2 μL, and the pulsed pressure injection mode was used for better sensitivity. Non-pulsed and pulsed detection with pressures of 10, 20, 35, and 40 psi were compared for the best peak area enhancement. The best peak areas resulted from 35 and 40 psi (Figure 1). However, there was no significant difference between the 35 and 40 psi injections; hence, 35 psi was chosen as the optimum pressure for pulsed injection. Figure 2 shows the chromatographic peaks of seventeen PIs; these were obtained with good resolution and peak shape.

### 3.3. Optimization of the Extraction Solution and Salts

Acetone, acetonitrile, and ethyl acetate are extraction solvents widely used for pesticide multiresidue analysis [25]. These solvents can be modified by changing the type of solvent or the analysis method [26]. QuEChERS is a pre-treatment method used for pesticides and other components [27,28]. After QuEChERS was introduced by Anastassiades et al. [15], acetonitrile was widely used as the extraction solvent in pesticide analysis. In addition, acetonitrile is compatible with both LC–MS and GC–MS. Analysis of pesticides in livestock, including high-fat samples, has been performed using the modified QuEChERS method [13,19,29,30]. Acetonitrile combined with other solvents has been studied for use in the extraction step [16]. 

The target compounds of the PIs in this experiment were non-polar compounds with log Pow ranges between 4.6 and 8.2. Therefore, the QuEChERS method was modified to optimize the pretreatment method for the experiment. To optimize the extraction solvent, acetonitrile, acetone, and ethyl acetate/acetonitrile (1/9, 2/8, and 3/7 *v*/*v*) were compared (Figure 3). Pork and eggs were chosen for comparison because of their high fat and protein content, respectively. The fortified concentration of the standard mixture was 0.01 mg/L. For acetonitrile extraction, 11 and 9 pesticides showed over 80% recovery from pork and eggs, respectively. For acetone extraction, 80% recovery was obtained for 12 and 10 pesticides in the pork and egg samples, respectively. Ethyl acetate was used as the extraction solvent to enhance the extraction efficiency of the non-polar pesticides. Using ethyl acetate/acetonitrile (1/9 *v*/*v*) extraction, 15 pesticides showed over 80% recovery from pork and eggs. When the ethyl acetate content was 20% and 30% in acetonitrile, 17 pesticides showed approximately 80% recovery from pork and eggs, respectively. As mentioned above, the log Pow range of the target PIs in this experiment was between 4.6 and 8.2. This indicates that the target compounds were non-polar. The polarity index of ethyl acetate is lower than that of acetone. Other studies used extraction solvents, including ethyl acetate, in livestock matrices [14,18]. These factors might have affected the results of the extraction solvent comparison. In addition, some studies have used extraction solvents combined with ethyl acetate to analyze pesticides in high-fat samples [18,31].

An extraction-salt comparison was performed. Original, AOAC, and EN salts were compared using the QuEChERS method. The original extraction salt is an unbuffered salt, which causes pH dependence during extraction. EN and AOAC are buffered salts that adjust the pH to approximately 5. The fortified concentration of the standard mixture was 0.01 mg/L. All the extraction salts demonstrated over 80% recovery. These results indicated that the PIs used in this experiment were not pH–dependent. Other studies have shown better recovery of pH–dependent pesticides using AOAC salts [27]. From the results of this study, it can be assumed that there were no significant differences in the pH of the sample extracts. Because all the extraction salts produced similar results, the original extraction salt was chosen, which is the simplest type, and was assumed to produce the least consequential matrix effect.

### 3.4. Optimization of d-SPE for Purification

Compared to SPE cartridge cleanup, d-SPE saves time and requires less solvent. Many studies have used d-SPE as a clean-up step in MS analysis [2,15]. Therefore, d-SPE was selected for the cleanup procedure. Different d-SPE compositions were compared, including combinations of PSA, C18, and GCB. PSA is suitable for the removal of organic acids and sugars. Graphitized carbon black (GCB) is widely used to remove pigments in samples, such as chlorophyll, and C18 is mainly used to improve the extraction efficiency of the pesticide in high-fat samples [32,33]. Pork was used for the comparison experiment, and the fortified concentration was 0.01 mg/L. All the d-SPE compositions showed over 80% recovery. The cleanup efficiency was tested by comparing the matrix effect based on the excellent extraction efficiency of all the d-SPEs. Unlike the results of other GC and LC detection types, the MS/MS results can fluctuate based on the sample type referred to as the matrix effect [34]. A matrix-matched standard was used for quantification to reduce the matrix effect. In this experiment, the slope of the calibration curve was used to calculate the matrix effect. As shown in Figure 4, d-SPE III (MgSO4 150 mg, PSA 25 mg, and GCB 25 mg) produced the lowest matrix effect; therefore, it was chosen for cleanup.

### 3.5. Limit of Quantitation (LOQ) and Calibration Curve of the Optimized Method

The matrix effect appears most often in MS analysis. The matrix effect enhances or suppresses the peak area due to interfering compounds in the sample extract [34]. This kind of enhancement or suppression can cause errors in the quantification of the target analyte. Although d-SPE cleanup reduces the matrix effect, using a matrix-matched calibration curve prevents errors in the quantification of the target compound.

The instrumental LOQs were all under 0.01 mg/L, and the linearity of the calibration was over 0.99. The method LOQ takes all analysis procedures into account. The instrumental LOQ, injection volume, sample quantity, and concentration ratio were calculated to determine the LOQ. In this experiment, the instrument LOQ was <20 pg, the injection volume was 2 μL, the extraction solvent volume was 20 mL, the sample weight was 5 g, and the concentration ratio was 1. Therefore, the LOQs were calculated to be 0.01 mg/L. In addition, the CV for all samples was less than 3% using seven replicates of a 0.02 mg/L matrix-matched standard solvent. The linearity of the PIs in the extracts of animal-derived foods was determined through linear regression of matrix-matched calibration curves in the 0.005–0.03 mg/L range. The coefficients of determination (R2) in beef were from 0.9949–0.9998 for beef, 0.9904–0.9995 for pork, 0.9923–0.9998 for chicken, 0.9934–0.9995 milk, and 0.9936–0.9995 for egg. The T coefficients of determination satisfied the criterion of >0.98.

### 3.6. Method Validation of the Established Method

The established method is illustrated in Figure 5. The accuracy and precision of the measurements were confirmed by conducting a recovery test using the established method. Five representative livestock samples were spiked with the standard solution. The fortified levels were 0.01 mg/L (LOQ), 0.1 mg/L (10LOQ), and 0.5 mg/L (50LOQ), with five replicates on each level. The recovery of beef was from 88.7–109.2%, 80.9–106.3% for pork, 75.4–108.8% for chicken, 80.4–109.8% for milk, and 75.2–106.2% for egg. The coefficient of variation for beef was from 1.2–8%, 0.4–6.9% for pork, 0.5–5.2% for chicken, 0.9–6.6% for milk, and 0.6–6.7% for egg. The recovery was 70–120%, and the coefficient of variation was less than 20% (Table 3). The matrix effects were in the range of −35.8 to 56.0%. These criteria were referred to SANTE/11945/2015 [20].

## 4. Conclusions

GC–MS/MS is suitable for trace-level analysis as it demonstrates high sensitivity and good reliability [6]. The QuEChERS pretreatment was found to be appropriate to accompany MS. By performing comparison experiments to optimize the QuEChERS method, the most suitable GC–MS/MS method for PIs was developed. Adding the concentration step to the pretreatment process led to a lower concentration of the target compound in the sample. Combining acetonitrile and ethyl acetate to make the extraction solvent led to optimized extraction efficiency. Ethyl acetate particularly improved the extraction efficiency. By comparing different compositions of d-SPE, the cleanup was optimized, and the matrix effect was reduced. Looking at the developed method as a whole, every step was improved, and the addition of concentration went beyond relying on instrument sensitivity alone. In addition, by using a mixture of two solvents rather than a single solvent in the extraction, the optimum efficiency was found according to the solvent combination rather than by comparing simple solvents. Method validation was performed on livestock samples based on the developed method and the data collected from the comparison experiment. All results satisfied the criteria referred to as SANTE/11945/2015 [20]. All data collected from this experiment is expected to be useful for further studies on pesticide analysis in livestock.

## Figures and Tables

**Figure 1 foods-11-03634-f001:**
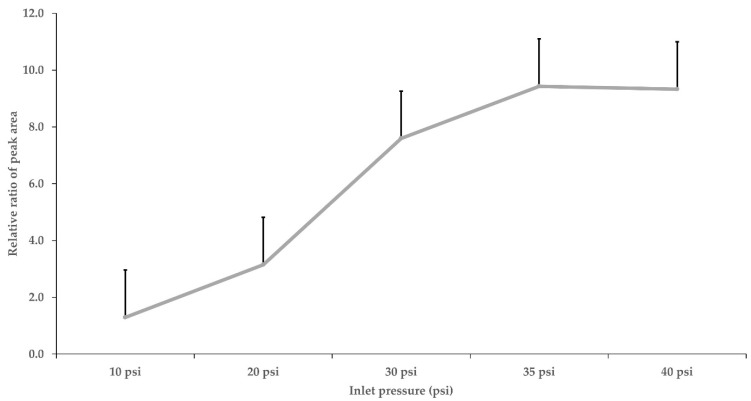
Effects of pulsed pressure during injection on the peak area of pyrethroid insecticides (0.01 mg/L). The Peak area from non-pulsed pressure injection was set to 1.

**Figure 2 foods-11-03634-f002:**
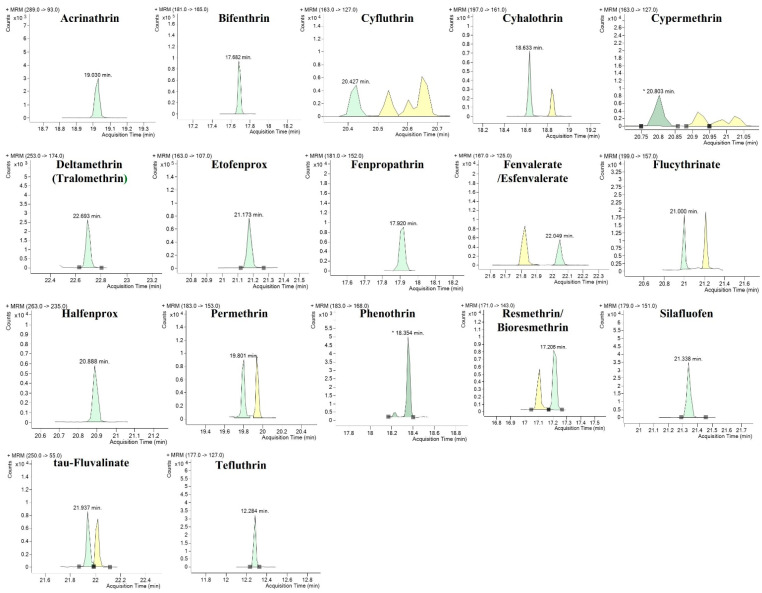
Multiple Reaction monitoring chromatograms of seventeen pyrethroid insecticides (0.02 mg/L).

**Figure 3 foods-11-03634-f003:**
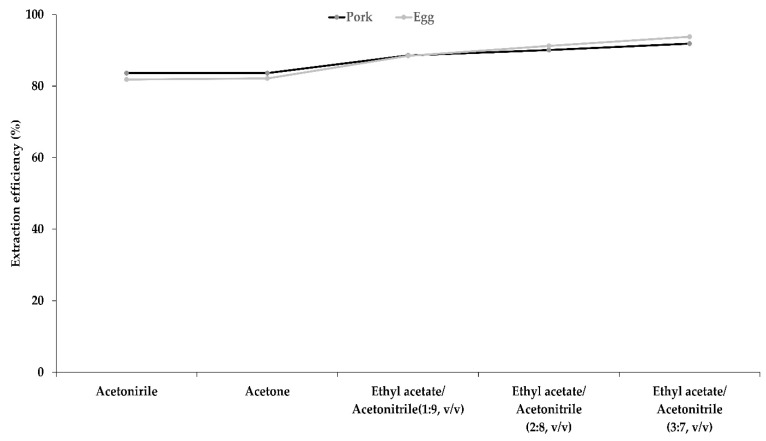
Extraction efficiencies of pyrethroid insecticides in pork and egg using various extraction solvents.

**Figure 4 foods-11-03634-f004:**
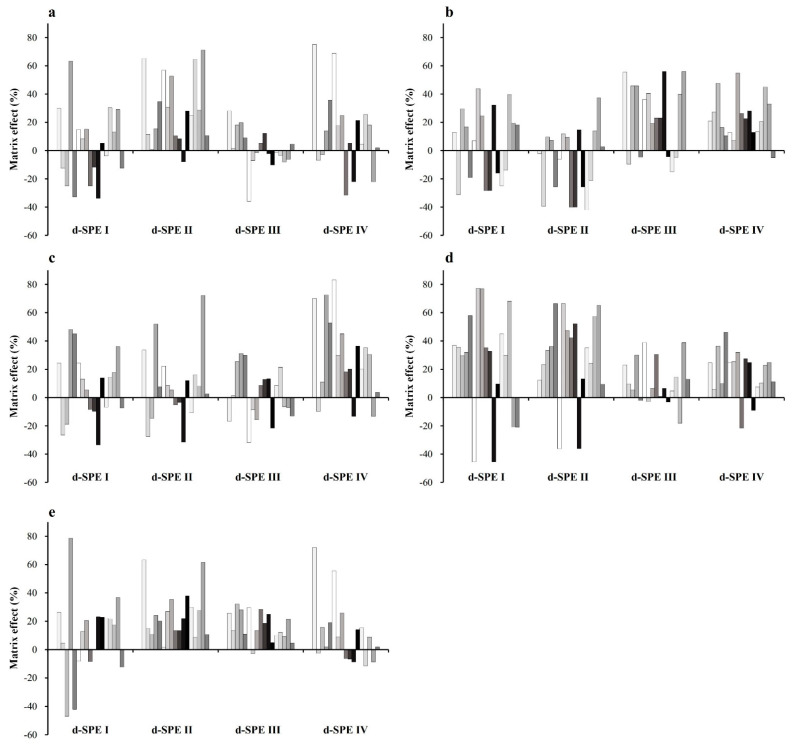
Matrix effects of pyrethroid insecticides in beef (**a**), pork (**b**), chicken (**c**), milk (**d**), and eggs (**e**) using dispersive solid phase extraction (d-SPE) mixtures I–IV. d-SPE I–MgSO_4_ 150 mg, primary secondary amine (PSA) 25 mg; d-SPE II–MgSO_4_ 150 mg, PSA 25 mg, C18 25 mg; d-SPE III–MgSO_4_ 150 mg, PSA 25 mg, graphitized carbon black (GCB) 25 mg; d-SPE IV–MgSO_4_ 150 mg, and C18 25 mg.

**Figure 5 foods-11-03634-f005:**
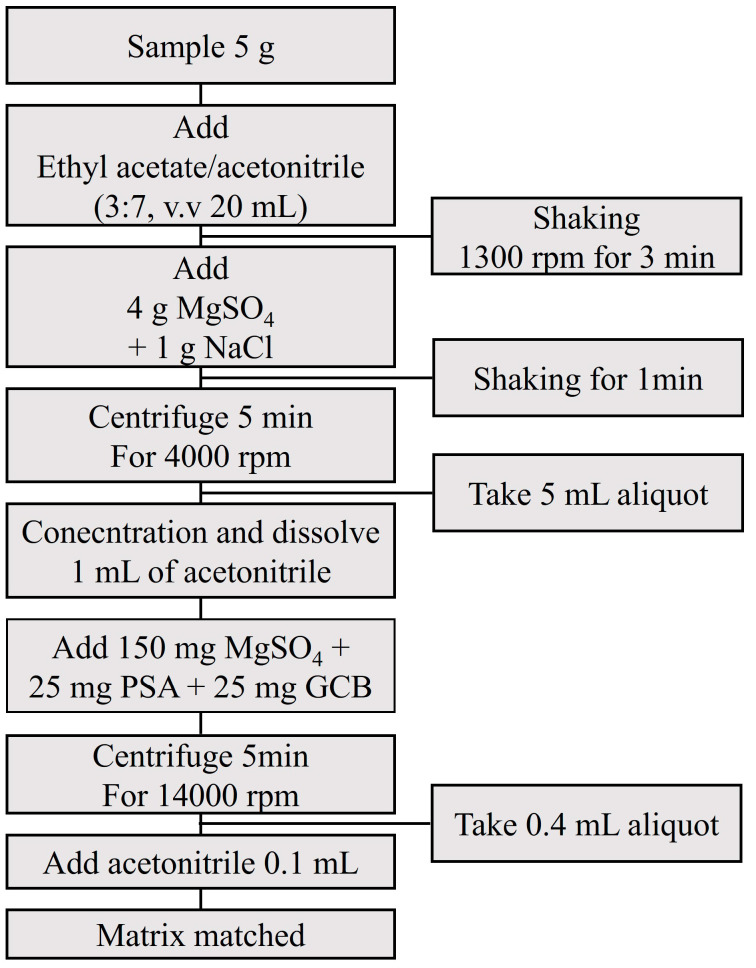
Established method of pyrethroid insecticide analysis.

**Table 1 foods-11-03634-t001:** Analytical methodologies applied for analyzing pesticides, including pyrethroid insecticide.

**Extraction Solvent**	**Pretreatment**	**Sentence**
**MeCN + 0.1% formic acid**	**QuEChERS + SPE**	**[11]**
**MeCN + 5% formic acid**	**EMR liquid clean up**	**[12]**
**MeCN/water (1:1, *v*/*v*)**	**QuEChERS + SPE**	**[13]**
**MeCH + 0.1% formic acid/ethyl acetate (7:3, *v/v*)**	**QuEChERS + dSPE**	**[14]**
**MeCN**	**QuEChERS + dSPE**	**[15]**
**MeCN**	**SPE**	**[16]**
**MeCN/ethoanol (95:5, *v*/*v*)**	**SPE**	**[17]**
**MeCN/ethyl acetate (1/1, *v*/*v*)**	**QuEChERS**	**[18]**
**acetone**	**QuEChERS + SPE**	**[19]**

**Table 2 foods-11-03634-t002:** GC–MS/MS condition for pyrethroid insecticides.

Compound	Molecular Weight	RT ^1^ (min)	Precursor Ion > Product Ion (CE ^2^, V ^3^)	
			Quantifier (m/z)	Qualifier (m/z)
Acrinathrin	541.4	19.0	289 < 93 (5)	289 < 77 (40)
Bifenthrin	422.9	17.6	181 < 206 (30)	181 < 115 (55)
Cyfluthrin	461.1	20.4, 20.5, 20.6, 20.6	226 < 206 (20)	163 < 127 (5)
Cyhalothrin	449.9	18.6, 18.8	197 < 161 (5)	197 < 141 (5)
Cypermethrin	416.3	20.8, 20.9, 20.9, 21.2	163 < 127 (5)	163 < 109 (30)
Deltamethrin(Tralomethrin)	381.1	22.7	253 < 174 (5)	253 < 172 (10)
Etofenprox	376.5	21.1	163 < 135 (10)	163 < 107 (20)
Fenpropathrin	349.4	17.9	265 < 210 (30)	181 < 152 (30)
Fenvalerate/Esfenvalerate	419.9	21.8, 22.0	225 < 119 (15)	167 < 125 (15)
Flucythrinate	451.5	21.0, 21.2	199 < 157 (5)	199 < 107 (15)
Halfenprox	477.3	20.9	263 < 235 (10)	263 < 169 (30)
Permethrin	391.3	19.8, 19.9	183 < 165 (20)	183 < 153 (20)
Phenothrin	350.5	18.3	183 < 168 (20)	183 < 153 (20)
Resmethrin/Bioresmethrin	338.4	17.1, 17.2	177 < 157 (10)	177 < 127 (25)
Silafluofen	408.6	21.3	286 < 258 (15)	179 < 151 (15)
tau-Fluvalinate	502.9	21.9, 22.0	250 < 200 (20)	250 < 55 (20)
Tefluthrin	418.7	12.3	177 < 157 (10)	177 < 127 (25)

RT ^1^: Retention time, CE ^2^: Collision energy, V ^3^: Voltage.

**Table 3 foods-11-03634-t003:** Recovery (n = 5) of pyrethroid insecticides in foods of animal origins using modified QuEChERS and GC–MS/MS.

Insecticide	Fortification	Recovery, %				
		Beef	Pork	Chicken	Milk	Eggs
Acrinathrin	LOQ	105.8 ± 3.0	94.0 ± 1.3	87.1 ± 5.0	109.8 ± 5.0	98.6 ± 3.6
	10LOQ	100.6 ± 6.1	83.8 ± 1.8	98.0 ± 1.3	92.0 ± 6.6	90.5 ± 4.5
	50LOQ	106.4 ± 4.5	94.9 ± 2.0	93.5 ± 2.0	101.9 ± 3.2	85.5 ± 1.2
Bifenthrin	LOQ	99.1 ± 1.9	101.3 ± 4.5	82.1 ± 1.6	100.1 ± 2.9	83.8 ± 5.7
	10LOQ	99.7 ± 4.0	89.8 ± 5.5	85.0 ± 1.5	90.3 ± 4.3	85.0 ± 1.0
	50LOQ	109.0 ± 2.0	82.7 ± 1.9	83.2 ± 0.8	90.9 ± 6.6	87.8 ± 2.5
Cyfluthrin	LOQ	105.3 ± 4.8	91.2 ± 2.6	90.8 ± 1.3	109.8 ± 3.3	90.8 ± 5.6
	10LOQ	100.9 ± 8.0	85.0 ± 1.2	90.6 ± 1.2	101.3 ± 4.8	90.9 ± 0.7
	50LOQ	104.1 ± 5.0	93.1 ± 1.7	88.5 ± 1.1	95.5 ± 4.4	84.8 ± 2.7
Cyhalothrin	LOQ	103.1 ± 2.8	93.3 ± 1.6	97.8 ± 3.2	107.5 ± 3.0	89.0 ± 5.9
	10LOQ	99.7 ± 7.4	86.1 ± 2.2	95.0 ± 1.6	91.6 ± 4.4	92.8 ± 1.2
	50LOQ	104.7 ± 5.4	94.5 ± 3.8	93.3 ± 2.0	84.5 ± 1.1	84.4 ± 4.4
Cypermethrin	LOQ	100.9 ± 4.9	88.9 ± 1.9	96.2 ± 3.9	104.5 ± 4.2	91.7 ± 0.6
	10LOQ	98.5 ± 6.8	81.8 ± 1.5	88.6 ± 0.5	93.0 ± 4.5	87.4 ± 1.6
	50LOQ	104.2 ± 3.4	90.4 ± 1.5	84.4 ± 0.8	91.8 ± 3.6	87.0 ± 1.2
Deltamethrin (Tralomethrin)	LOQ	90.6 ± 3.6	91.1 ± 3.0	90.4 ± 3.6	106.7 ± 1.2	93.2 ± 2.7
	LOQ	88.7 ± 1.6	82.1 ± 2.0	88.7 ± 1.6	88.6 ± 5.6	93.0 ± 2.3
	10LOQ	98.7 ± 1.8	91.3 ± 1.6	98.7 ± 1.8	94.5 ± 4.4	89.4 ± 4.9
Etofenprox	LOQ	98.9 ± 2.8	84.6 ± 3.3	80.4 ± 2.6	92.7 ± 1.4	88.0 ± 3.0
	10LOQ	97.5 ± 7.9	87.0 ± 5.1	94.8 ± 1.0	83.2 ± 3.3	82.2 ± 1.0
	50LOQ	104.1 ± 2.7	89.4 ± 3.7	94.3 ± 2.6	91.0 ± 1.9	85.0 ± 3.2
Fenpropathrin	LOQ	106.9 ± 3.9	88.9 ± 0.4	90.0 ± 2.3	105.2 ± 5.9	94.5 ± 3.1
	10LOQ	102.2 ± 5.8	81.7 ± 0.9	90.6 ± 3.6	91.3 ± 3.9	88.4 ± 1,0
	50LOQ	108.1 ± 3.1	90.3 ± 1.7	86.4 ± 2.9	91.6 ± 4.2	87.8 ± 3.5
Fenvalerate/ Esfenvalerate	LOQ	100.9 ± 1.2	91.6 ± 1.2	82.3 ± 4.0	106.1 ± 3.2	93.4 ± 6.6
	10LOQ	100.0 ± 6.3	84.7 ± 3.9	92.9 ± 3.0	92.5 ± 3.3	84.1 ± 3.9
	50LOQ	108.6 ± 2.0	88.8 ± 2.2	86.1 ± 1.8	96.8 ± 1.9	83.1 ± 2.1
Flucythrinate	LOQ	98.0 ± 3.4	94.5 ± 3.7	97.4 ± 4.9	101.2 ± 2.4	101.7 ± 6.4
	10LOQ	104.8 ± 7.8	87.1 ± 1.7	94.9 ± 2.7	97.2 ± 4.5	96.4 ± 4.8
	50LOQ	104.2 ± 3.1	89.4 ± 3.3	100.9 ± 4.2	102.0 ± 2.3	84.9 ± 2.6
Halfenprox	LOQ	101.0 ± 4.2	104.8 ± 2.5	84.5 ± 2.7	92.7 ± 4.7	83.4 ± 3.7
	10LOQ	96.9 ± 3.4	88.6 ± 4.5	83.4 ± 1.8	84.5 ± 1.7	83.0 ± 2.6
	50LOQ	101.7 ± 3.9	99.9 ± 2.2	87.0 ± 3.6	92.0 ± 4.1	88.5 ± 2.2
Permethrin	LOQ	98.8 ± 4.5	83.7 ± 3.0	84.5 ± 5.2	98.9 ± 4.4	85.9 ± 3.0
	10LOQ	100.1 ± 4.6	84.3 ± 3.1	78.0 ± 4.9	88.8 ± 2.8	89.4 ± 3.1
	50LOQ	95.2 ± 6.4	87.6 ± 3.8	75.4 ± 1.0	83.0 ± 0.9	80.9 ± 1.5
Phenothrin	LOQ	102.2 ± 1.9	104.8 ± 2.2	108.8 ± 5.1	108.8 ± 5.1	87.3 ± 2.1
	10LOQ	100.3 ± 5.6	82.8 ± 2.3	93.0 ± 3.8	93.0 ± 3.8	84.4 ± 2.6
	50LOQ	101.8 ± 3.5	85.3 ± 3.8	89.6 ± 4.3	89.6 ± 4.3	88.9 ± 1.6
Resmethrin/ Bioresmethrin	LOQ	94.8 ± 3.7	87.8 ± 3.4	106.4 ± 4.3	97.6 ± 2.2	92.4 ± 5.3
	10LOQ	91.4 ± 5.8	92.0 ± 0.9	102.2 ± 2.9	80.5 ± 1.7	96.3 ± 6.7
	50LOQ	107.1 ± 3.4	89.0 ± 2.5	98.0 ± 2.5	92.8 ± 3.3	81.1 ± 2.0
Silafluofen	LOQ	93.9 ± 1.7	106.3 ± 1.4	83.3 ± 3.5	93.9 ± 3.1	84.5 ± 5.0
	10LOQ	93.9 ± 4.3	80.9 ± 3.7	81.1 ± 3.4	80.4 ± 2.1	88.1 ± 3.7
	50LOQ	100.9 ± 2.5	93.9 ± 6.9	82.2 ± 2.3	87.8 ± 2.3	85.8 ± 4.0
tau-Fluvalinate	LOQ	103.5 ± 2.2	92.4 ± 1.7	83.9 ± 2.3	109.3 ± 3.5	106.2 ± 3.2
	10LOQ	101.3 ± 5.4	83.7 ± 0.9	87.5 ± 1.6	92.8 ± 3.7	94.1 ± 1.9
	50LOQ	105.9 ± 5.3	97.6 ± 2.3	84.3 ± 1.1	90.6 ± 1.6	92.5 ± 4.5
Tefluthrin	LOQ	105.2 ± 2.0	88.9 ± 1.1	91.5 ± 1.7	103.7 ± 2.1	87.0 ± 2.0
	10LOQ	104.0 ± 2.3	82.8 ± 0.6	88.1 ± 1.6	100.2 ± 3.9	89.0 ± 0.7
	50LOQ	109.2 ± 2.1	87.9 ± 2.8	86.9 ± 1.3	101.5 ± 4.7	75.2 ± 1.0

## Data Availability

Data is contained within the article.

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
