# Peer review of "Simultaneous Determination of Pyrethroid Insecticides in Foods of Animal Origins Using the Modified QuEChERS Method and Gas Chromatography-Mass Spectrometry"

_foods, 2022, doi:10.3390/foods11223634_

Round 1
Reviewer 1 Report
The present manuscript entitled "Simultaneous Determination of Pyrethroid Insecticides in Foods of Animal Origins using Modified QuEChERS method and Gas Chromatography-Mass Spectrometry" by Byung Joon Kim, Seung-Hyun Yang, and Hoon Choi (foods-1983136) is written correctly and has a good structure; moreover, it has all the necessary parts. The article is interesting from an analytical and food analysis point of view; therefore, it should interest the reader. I proposed improvements in the method description and with a presentation of figures. My current decision is a minor revision. More specific comments and observations are presented below.
1. Please correct typos, especially indexes in compounds or abbreviations of parameters.
2. RSD expressed as a percentage is the coefficient of variation (CV).
3. Figure 1. Shouldn't the range also appear below the mean value?
4. Figure 2. Is it possible to export the data and prepare the drawing in another program of better quality?
5. The authors mention the matrix effect due to interferences. What can be done in the event of strong interference effects? How would you deal with them? What types of interference effects could occur?
6. Page 10, line 262. Shouldn't the unit be converted to pg/mL?
7. Table 2 should be better commented in the text.
8. Does the developed method have disadvantages?
9. Conclusion. Please, emphasize clearly the advantages of the research carried out.
10. Appropriate tools should be used to best characterize the method when developing a new approach (e.g., RGB Additive Color Model to Analytical Method Evaluation, AGREE, or GAPI).
11. Page 12, lines 297-298. SI is mentioned, but no access.
12. References. Please adjust to journal requirements.
I hope that the comments presented will help improve the article.
Author Response
Thank you for your all the comment. Pleas see the attachment

Reviewer 2 Report
The aim of this study is to provide a simultaneous residue analysis method for quantitative analysis of seventeen pyrethroid insecticides in foods of animal origin, including beef, pork, chicken, milk, and eggs. The arrangement of experimental procedures is appropriate. But the innovation of this manuscript is insufficient, and some comments should be addressed by authors.
(1) Many similar research works have been reported, this paper does not highlight its innovative power or differences.
(2) In the introduction section, the current status of pyrethroid insecticides detection skills has been described too little, and it has not been explained why the detection method used in this research work, the advantages?
(3) It is recommended to use a line chart for the data in Figure 1, with only positive errors and no negative error lines.
(4) The gas chromatography-mass spectrometry of seventeen pyrethroid insecticides standard should be provided in the manuscript.
(5) It is recommended to use a line chart for the data in Figure 3, and the extraction efficiencies has no obvious change after using various extraction solvents, why not consider other proportions of ethyl acetate/acetonitrile (4:6, 5:5….).\
(6) Are the LOD of different samples the same? The extraction efficiency of different samples is different, so the LOD should be different.
(7) The results of this paper lack the standard curves of the detection methods for seventeen pyrethroid insecticides of different varieties.
Author Response
Thank you for your comment. I tried my best to answer your comment below. But if you want more answer or answer is no sufficient, let me have another chance to answer. Please see the attachment file below for my answer for your comment.

Reviewer 3 Report
The Manuscript entitled “Simultaneous Determination of Pyrethroid Insecticides in Foods of Animal Origins using Modified QuEChERS method and Gas Chromatography-Mass Spectrometry” has been reviewed and I report some minor considerations and suggestion as follows:
/Authors report a detailed manuscript concerning the screening of pesticides in food products. The analytical methods have been well described, correctly developed from extraction procedures to gas chromatographic/mass spectrometric determination. In my opinion, this paper can be considered by other Authors for the determination of pesticides in food context. In opinion of this Reviewer, this research article can be published on the FOODS, previous minor modifications. For this reason, this Reviewer suggests a minor revision and some comments are listed below:/
//
1. /Are there mrl for target pesticides in foods products? It is
possible identify and quantify target compounds at mrl levels?/
2. /Why do the Authors use 2015 Sante guidelines? In my opinion, it is
best to report the current Sante guideline. /
3. /Line 85: please, correct Restock./
Author Response
Thank you for your comment. And comment were pointing the part I was missing. I tried my best to answer. If you want some more answer or answer was sufficient, please let me answer once again. Please see the attachment below.
Thank you

Reviewer 4 Report
1) MLOQ and MLOD abbreviations from the beginning to the end of the article should be corrected as LOQ and LOD.
2) R2 should be written instead of r2 in the abstract.
3) A table should be added by comparing previous studies for the determination of Parathyroid inseticides. Especially in a way that demonstrates the superiority of your method.
4) Concentration units must be in a single format. Either mg/L or μg/L should be preferred.
5) Table 1 should be rearranged according to significant figures.
6) The grammatical structure of the article should be checked and it should be in fluent English.
Author Response
Thank you for your comment. I hope my answer for your comment is sufficient. If my answer is not enough or need more other comment, please let me do it again.

Round 2
Reviewer 2 Report
no.